# OpenReview forum: "Benchmarking the Robustness of Agentic Systems to Adversarially-Induced Harms"
_ICLR.cc/2026/Conference — Submitted to ICLR 2026_

### Official Review · Reviewer_dMaP · 2025-10-29

**Soundness:** 2
**Presentation:** 2
**Contribution:** 2
**Rating:** 4
**Confidence:** 5

**Summary:**

This paper introduces BAD-ACTS, a benchmark for testing LLM-based multi-agent system security where one compromised agent manipulates others into harmful actions. It includes a taxonomy of 17 harm types, 5 environments with different architectures, and 937 curated attack examples. Testing 8 models (Llama, GPT, Mistral, Qwen), results show all are vulnerable (28-53% attack success), larger models are paradoxically more vulnerable, centralized architectures are safer than decentralized, and models defend poorly against agent-specific threats (denial-of-service: 50%, resource stealing: 67%) versus traditional harms (toxicity: 31%). Two defenses are proposed: Adversary-Aware Prompting (ineffective) and Guardian Agents (26-63% ASR reduction), demonstrating that current LLM safety training doesn't generalize to multi-agent scenarios.

**Strengths:**

1. Uses exact tool call and parameter matching to calculate ASR, providing more accurate and reproducible evaluation than LLM-as-judge approaches (98% vs 80% human agreement)
2. Comprehensive taxonomy systematically categorizes 17 sub-types of agentic harms across malware generation, malicious human interactions, harmful content, and unauthorized actions, establishing a foundational framework for multi-agent security research
3. High-quality curated dataset with 238 manually-reviewed core examples and 699 extended examples spanning 5 realistic environments, ensuring diversity and quality control through iterative LLM generation with human curation.
4. Systematic evaluation across diverse communication architectures (decentralized, centralized, hierarchical, sequential) reveals important architectural security implications, showing centralized/hierarchical structures are inherently more robust against adversarial manipulation than decentralized designs

**Weaknesses:**

1. The paper exclusively focuses on attack success rate without measuring how defenses impact normal task completion or benign performance, providing no evaluation of security-utility tradeoffs which is critical for assessing whether Guardian Agents' ASR reduction (26-63%) comes at acceptable costs to system functionality in practical deployments.
2. Only two basic detection-based defenses are tested while established techniques from recent literature—such as StruQ[1] for structured query filtering and adversarially-trained defense mechanisms[2]—are neither evaluated nor discussed, limiting the comprehensiveness of the defensive baseline and missing opportunities to compare against systematic resistance mechanisms directly relevant to multi-agent manipulation attacks.
3. The paper lacks concrete technical implementation details including message passing mechanisms, turn-taking control logic, tool call parsing from LLM outputs, state management, and LLM API integration, providing no code snippets, algorithms, or architectural diagrams which makes reproduction difficult and leaves potential implementation-specific vulnerabilities unexplored.
4. While the paper provides example attack episodes in Appendix G, it lacks in-depth qualitative case studies analyzing why specific attacks succeed or fail, what persuasion strategies adversaries employ, and how different models' reasoning processes lead to vulnerabilities. Such case studies could be summarized using short “Takeaway” paragraphs inside the main text to highlight practical insights about both attack patterns and potential defense signals.

[1] Chen et al., *StruQ: Defending Against Prompt Injection with Structured Queries*, USENIX Security 2025.

[2] Chen et al., *Defense Against Prompt Injection Attack by Leveraging Attack Techniques*, ACL 2025.

**Questions:**

1. Several agent security benchmarks already exist, such as Agent Security Bench (ASB) [1] and AgentDojo [2], yet the paper does not provide a comparative analysis distinguishing BAD-ACTS in terms of scope, methodology, threat model, or attack coverage—a comparison table clarifying how this benchmark's multi-agent manipulation focus, communication structure diversity, and harm taxonomy differ from existing single-agent security benchmarks would strengthen the positioning.
2. The paper does not explain how tools and their functionalities were implemented, whether they were adapted from existing agent frameworks (AutoGen, LangChain) or custom-designed.
3. Table 2 contains a typographical error where 'ARS' should be 'ASR' (Attack Success Rate) to maintain consistency with the paper's terminology throughout.
4. The paper does not test popular production multi-agent systems such as Claude Code or Codex, instead evaluating only individual LLM models (Llama, GPT, etc.) in custom-built simulated environments—testing actual deployed systems would provide stronger ecological validity, though the authors' choice of simulation is justified for safety and reproducibility, this limitation should be explicitly acknowledged as it affects real-world applicability of the findings.

[1] Zhang et al., Agent Security Bench (ASB): Formalizing and Benchmarking Attacks and Defenses in LLM-based Agents, ICLR 2025.

[2] Debenedetti et al., AgentDojo: A Dynamic Environment to Evaluate Prompt Injection Attacks and Defenses for LLM Agents, NeurIPS 2024.

---

> ### Author Response · Authors · 2025-11-19
> **Response to Reviewer dMaP**
>
> We thank the Reviewer for their valuable feedback. We are delighted to read that the reviewer appreciates our evaluation method, our taxonomy, dataset and the evaluation. We will address their feedback in the following:
>
> ### Weaknesses
> 1. We thank the reviewer for their suggestion, and agree that measuring the impact of defense mechanisms on regular benign episodes beyond the reported false-positive rate is a useful addition. To this end, we evaluated the Quality of a plan in the Travel-Planning environment using an LLM-Judge, where we found no significant differences between the setting with no defense (Quality=3.34/5), Adversary-Aware Prompts (Quality=3.35/5), and Guardian-Agents(Quality=3.40/5). We observed the same in the Financial Article Writing environment (No Defense : 3.67/5, Adversary-Aware-Prompts : 3.70/5, Guardian Agents : 3.70/5). This demonstrates that the defense methods do not have any negative impact on task completion. We will expand these experiments to all environments in future iterations of the paper.
>
> 2. We thank the reviewer for bringing these works to our attention. However, we believe that the methods cited are primarily designed to mitigate prompt injection attacks, which differ from the attack scenarios studied in our benchmark. Our focus is on evaluating defenses within the context of multi-agent manipulation and harmful behaviors, where these techniques are not directly applicable. We will clarify this distinction in the revised manuscript to avoid potential confusion regarding the scope of our defense evaluation.
>
> 3. We utilize the AutoGen library for most implementations of the mentioned aspects, while the turn-taking control logic depends on the communication structure illustrated in Figure 2. Additional details about the tools can be found in Appendix E. We respectfully disagree with concerns regarding reproducibility: all code will be publicly released, including in the supplementary material of this submission, and is thoroughly documented to allow replication of our experiments.
>
> 4. We thank the reviewer for this suggestion. We found that adversarial behaviors are highly diverse, making concise summarization challenging. Broadly, the patterns depend on both the model and agent: some adversaries simply instruct other agents to perform the harmful action, others engage in deceptive behaviors such as impersonation, and still others execute multi-step attacks, building trust by following their regular role before initiating the adversarial action. We plan to include these observations in future iterations.
>
> ### Questions
> 1. We thank the reviewer for this suggestion and agree that a comparative table would help clarify our benchmark’s positioning. BAD-ACTS primarily differs from existing benchmarks such as ASB and AgentDojo in its multi-agent focus, capturing attacks that emerge through interactions among multiple agents rather than single-agent settings. Additionally, our benchmark emphasizes diverse communication structures and a comprehensive harm taxonomy, which together enable systematic evaluation of the robustness of agentic systems. We have included a comparison table in Appendix C, and we will include this table within the introduction section of future revisions.
>
> 2. We thank the reviewer for raising this point. All tools in our benchmark are custom-designed to achieve a relevant level of fidelity while maintaining a lightweight and secure evaluation environment for agentic systems. Detailed descriptions of the tools and their functionalities can be found in Appendix E (Table 10). We will clarify this in the main text to ensure readers understand our design choices.
>
> 3. We thank the reviewer for pointing this error out, and we have fixed this.
>
> 4. We thank the reviewer for this suggestion. We believe that our code generation environment aligns closely with existing systems and our implementation allows for a more standardized evaluation. We will clarify this in future revisions.

---

### Official Review · Reviewer_fuFn · 2025-10-29

**Soundness:** 3
**Presentation:** 3
**Contribution:** 2
**Rating:** 6
**Confidence:** 2

**Summary:**

The authors introduce BAD-ACTS, a benchmark and taxonomy on evaluating multi-agent systems under adversarial manipulation. The authors implement five environments (Travel Planning, Personal Assistant, Financial Writing, Code Generation, Multi-Agent Debate) with emulated tools and inter-agent communication graphs. They evaluate on 900+ adversarial actions.

**Strengths:**

- Timely & impactful problem framing and taxonomy.
- Diverse, well-scoped environments.
- Clearly written.

**Weaknesses:**

- **Threat model:** The primary attack assumes a fully compromised agent with role-conformant messaging. How do results translate to more realistic threat models (e.g., partial prompt corruption, compromised tool output, or IPI). Any treatment of weaker adversaries would strengthen claims about general robustness.
- **Evaluation metric:** Keyword metrics could under-count semantically successful but lexically different attacks and over-count near-misses. More detail on failure modes of the keyword evaluation would help.
- **Tools:** Tools are emulated, but side effects can have material impact on ASR. Any discussion or ablation on the real-world risks that were abstracted away would be nice.
- **Baselines:** It would be helpful to compare Guardian Agents against other defenses (e.g., least-privilege tool gating, simple rate-limiting).
- **Taxonomy:** The categories are sometimes scattered and mix harm mechanisms (e.g., file deletion) with content harms (e.g., toxicity). "Privacy" as a category could refer to a very broad set of threats (but seems to be used narrowly here?). Revisiting some of the verbiage here could be helpful.

**Questions:**

- Did you test weaker adversaries?
- Can you provide per-category precision/recall for the keyword evaluator vs. human? Any notable failure patterns?
- How sensitive are your results to prompt lengths, system prompt wording?

---

> ### Author Response · Authors · 2025-11-19
> **Response to Reviewer fuFn**
>
> We thank the reviewer for their valuable feedback. We are delighted to read that they appreciate the relevance of the considered problem, the proposed environments, and our writing. Regarding their feedback:
> ### Weaknesses
> 1. (and Q1) We appreciate the reviewer’s concern regarding the practical realism of our threat model. We believe that our considered threat setting is realistic. Individual agents may be compromised through existing mechanisms such as data poisoning or prompt injection attacks, and emerging research on agent marketplaces [Wang et. al, 2025] highlights scenarios where agents can be bought and sold, creating opportunities for adversarial deployment. Importantly, the experiments in this paper serve as a proof-of-concept, designed to be simple yet illustrative, to demonstrate the utility of our benchmark for evaluating robustness of agentic systems. The primary contribution of our work lies in the benchmark itself, rather than the novelty or sophistication of a particular threat, and this controlled threat setting provides a clear and consistent environment for systematic evaluation.
> However, we do agree with the reviewer that an evaluation of a more diverse set of threat models would be beneficial for the strength of the claims. We therefore conducted additional experiments, where we consider adversaries that are able to conduct a Prompt Injection in the user’s initial instruction (Appendix B.2) and manipulate the tools in the system (Appendix B.3). We found that these adversaries are still able to conduct effective attacks.
>
> 2. We thank the reviewer for raising this point. Our evaluation closely aligns with ground-truth outcomes, as evidenced by the high agreement with human evaluation (line 305). This indicates that the keyword metrics reliably capture the relevant attacks in our benchmark, though we acknowledge that more nuanced evaluation methods could be explored in future work for different kinds of adversarial actions then the ones considered here.
>
> 3. We believe that the emulation of the tools that we consider here is sufficiently realistic for a responsible security evaluation, as we imitate API-functionality without actually executing any, potentially harmful, action. Our evaluation follows the practices of prior works [1]. We will clarify this aspect.
>
> 4. We thank the reviewer for the suggestion. As noted in Line 260, we already implement least-privilege tool gating even in the environment without any defense mechanisms. Regarding simple rate-limiting, it would not prevent the types of attacks studied in our benchmark, as we consider single adversarial actions per iteration.
>
> 5. We thank the reviewer for this feedback and would appreciate any further clarification regarding the mixing of harm mechanisms and content harms. To clarify, actions such as deletion of private files and generation of toxic content (e.g., in generated newspaper articles) are both examples of harmful behaviors that agentic systems might exhibit, even if they manifest differently. We however agree that the category “Privacy” could be confusing, as multiple types of threats (e.g., release of private information) fall under this umbrella. To improve clarity, we will rename this category to “Plan Leakage” in the revised version.
>
> ### Questions
> 2. We thank the reviewer for the suggestion. In most categories, the keyword-based evaluation aligns perfectly with ground-truth labels, with all misclassifications occurring in Toxicity (Recall = 0.96) and Stealing Resources (Recall = 0.92). Notably, because all adversarial actions require outputting specific phrases, the precision is 1 across all categories. These results indicate that the keyword evaluator is highly reliable, with only minor recall limitations in a small subset of categories.
>
> 3. We thank the reviewer for raising this question. We found that choosing the correct system prompt is critical for developing benign agents capable of reliable tool use. In contrast, adversarial agents are generally simpler to implement, as we were able to use very similar system prompts across different environments and agents. That said, some models, particularly smaller ones, struggled with the long-context nature of agentic tasks, highlighting that performance can be sensitive to both prompt design and model capabilities.
>
> [1] Lynch, Aengus, et al. "Agentic Misalignment: How LLMs Could Be Insider Threats." arXiv preprint arXiv:2510.05179 (2025).

---

### Official Review · Reviewer_e46v · 2025-10-31

**Soundness:** 3
**Presentation:** 3
**Contribution:** 3
**Rating:** 6
**Confidence:** 4

**Summary:**

This paper introduces BAD-ACTS, a new benchmark for evaluating the robustness of Multi-Agent Systems (MAS) against a specific, internal threat model. The paper formalizes a scenario where one agent within a collaborative system is compromised and attempts to manipulate its peer agents into executing harmful, action-oriented tasks. The benchmark consists of five distinct MAS environments with varied communication topologies, a new taxonomy of "agentic harms," and a curated dataset of harmful actions. Using this benchmark, the authors evaluate several popular LLMs and find that this internal attack vector is effective. The results yield several interesting findings, such as larger models being more vulnerable to this form of manipulation.

**Strengths:**

1. **Novel Problem Formulation**: The paper's primary strength is its focus on a novel and important, if under-explored, threat model: internal adversarial manipulation within an MAS. It formalizes the "insider threat" problem for agentic systems, moving beyond typical external attacks (e.g., user jailbreaks).

2. **Comprehensive Benchmark Engineering**: The creation of five distinct environments with different communication structures (centralized and hierarchical) is a significant engineering effort. This design wisely allows for a more nuanced analysis of how system topology impacts security.

3. **Interesting and Non-Obvious Findings**: The experimental results provide valuable insights. The discovery that larger, more capable models (e.g., Llama-70b, GPT-4.1) are more susceptible to manipulation than their smaller counterparts is a significant and counter-intuitive finding for the community.

**Weaknesses:**

1. **Questionable Practicality of the Threat Model**: A noteworthy limitation is the practical realism of the threat model. The benchmark assumes an adversary has already achieved full control over one agent. The paper does not justify how this level of control is realistically achieved. Therefore, while the high ASRs are alarming, they are contingent on this "best-case" scenario for the attacker, which may not be broadly applicable.

2. **Lack of Rigor in Taxonomy Generation**: The novelty and rigor of the proposed taxonomy (Table 1) are unclear. The paper states it is based on a review of existing literature and the authors' "own analysis" [line 161] but fails to detail what this analysis entailed or its methodology. The taxonomy appears to be a useful compilation of harms already identified in prior work, rather than a novel, principled classification.

3. **Outdated Literature Context**: The paper's framing relies heavily on foundational MAS work from 2023. While appropriate for context, the field of agentic systems has evolved extremely rapidly. The lack of deeper engagement with more recent 2024/2025 literature on agent security and architecture makes the work feel slightly less current than its peers.

**Questions:**

Same with weaknesses above.

---

> ### Author Response · Authors · 2025-11-19
> **Response to Reviewer e46v**
>
> We thank the reviewer for their valuable feedback. We are delighted to read that they found our considered problem novel and important, they appreciate the amount of engineering, and that they found our findings interesting. In the following, we would like to address their concerns.
>
> 1. We appreciate the reviewer’s concern regarding the practical realism of our threat model. We believe our setting is plausible in real-world multi-agent scenarios. Individual agents may be compromised through existing mechanisms such as data poisoning or prompt injection attacks, and emerging research on agent marketplaces [1] highlights scenarios where agents can be bought and sold, creating opportunities for adversarial deployment. Importantly, the experiments in this paper serve as a proof-of-concept, designed to be simple yet illustrative, to demonstrate the utility of our benchmark for evaluating robustness of agentic systems. The primary contribution of our work lies in the benchmark itself, rather than the novelty or sophistication of a particular threat, and this controlled threat setting provides a clear and consistent environment for systematic evaluation.
> However, to increase the diversity of considered threat settings, we extended our experiments to include adversaries that are able to conduct Prompt Injections in the user’s initial request (Appendix B.2.) and manipulate the output of the used tools (Appendix B.3.). We generally found that attacks still remain effective.
>
> 2. We thank the reviewer for pointing out this inconsistency in our current writeup. The methodology behind our analysis of systems for the taxonomy was two-fold. First, we conducted a detailed analysis of existing tools commonly used in agentic systems and examined how they are typically misused by human adversaries. Second, we extended the taxonomy to incorporate any adversarial actions observed in our dataset generation that did not fit neatly into existing categories. This process allowed us to both systematically compile known harms and identify novel behaviors. We will move these methodological details from Appendix D.1 to the main text to make the rigor and novelty of our taxonomy clearer.
>
> 3. We thank the reviewer for highlighting this point. While our framing relies on foundational MAS work from 2023, we believe that these works serve as a foundation for a stable evaluation framework that remains relevant even as the field of agentic systems evolves rapidly. At the same time, we acknowledge the importance of engaging with the most recent literature, and we plan to incorporate additional 2024–2025 work on agent security, such as AgentHarm [2] or AFlow [3].
>
>
>
> [1] Wang, Yuntao, et al. "Internet of agents: Fundamentals, applications, and challenges." arXiv preprint arXiv:2505.07176 (2025).
>
> [2] Andriushchenko, Maksym, et al. "AgentHarm: A Benchmark for Measuring Harmfulness of LLM Agents." The Thirteenth International Conference on Learning Representations.
>
> [3] Zhang, Jiayi, et al. "AFlow: Automating Agentic Workflow Generation." The Thirteenth International Conference on Learning Representations.

---

> > ### Comment · Reviewer_e46v · 2025-11-27
> >
> > Thank you for the response. My primary concern remains the validity and logic of the proposed taxonomy. As noted by other reviewers, the categorization lacks novelty. Furthermore, the taxonomy appears to be a flat list derived from observation rather than a principled framework. The categories lack strong internal coherence or structural relationships. For example, the distinction between "Release Private Information" (categorized under Malware) and "Plan Leakage/Privacy" (categorized under Malicious Interaction) is ambiguous and lacks theoretical justification. Consequently, the theoretical contribution of the benchmark remains limited.

---

### Official Review · Reviewer_867L · 2025-11-01

**Soundness:** 2
**Presentation:** 1
**Contribution:** 2
**Rating:** 2
**Confidence:** 4

**Summary:**

This paper introduces BAD-ACTS, a benchmark for evaluating the security of LLM-based agentic systems against adversarial manipulation. The authors propose a taxonomy of potential harms specific to agentic systems and implement five diverse environments with different communication structures. The benchmark includes a dataset of 238 harmful actions across multiple categories. The authors evaluate a threat model where a single adversarial agent attempts to manipulate other agents into performing harmful actions, testing this across multiple state-of-the-art LLMs. Results show high attack success rates, with larger models often being more vulnerable. The paper also proposes two baseline defenses: adversary-aware prompting and guardian agents.

**Strengths:**

- 238 high-quality examples of harmful actions.
- Baseline defenses evaluated, such as safety-inducing prompts and zero-shot monitoring methods.

**Weaknesses:**

- The proposed taxonomy, framed as one of the key contributions, doesn’t list novel points. All of them have been considered in the previous literature.
- Heavy reliance on LLM generation for the dataset creation may have introduced potential biases or lack of coverage. It would be useful to expand a discussion on the realism of the tasks and tools.
- The proposed benchmark is not properly compared to existing benchmarks. For example, AgentHarm from ICLR 2025 is not discussed. Also, it would be good to discuss AgentDojo from NeurIPS 2024 in more detail.
- The claim *“We analyzed these results in more detail and found that larger models are more vulnerable to these attacks than smaller instances of the same family”* seems to be based only on the comparison of *“Llama-8B vs. 70B and GPT-o4-mini vs. GPT-4.1”*. It’s not clear whether o4-mini is smaller than GPT-4.1. Then the only reliable piece of evidence is Llama-8B vs. 70B which is insufficient to draw confident conclusions.
- The proposed defense with a zero-shot monitor works well, but this is likely due to the fact that the adversarial agent was not prompted to conceal its message from the monitor.

Moreover, the presentation of the benchmark can be improved:
- It would be useful to have some representative examples in the main part (beyond the single example from Figure 1). This is important, since the quality of examples is the main feature of any benchmark (especially, an agentic one).
- It would be more natural to describe the focus of the benchmark as “multi-agent systems” instead of “agentic systems”. The multi-agent part seems key in the proposed benchmark and is worth emphasizing.

**Questions:**

How realistic is the considered multi-agent threat model? In which cases can we expect a malicious agent in a multi-agent system?

---

> ### Author Response · Authors · 2025-11-19
> **Response to Reviewer 867L**
>
> We thank the reviewer for their helpful feedback. We are delighted to hear that they appreciate the proposed dataset and the inclusion of baseline defenses. We would like to address their constructive feedback in the following:
>
> ### Weaknesses
> 1. We appreciate the reviewer’s comment and indeed, several components of our taxonomy build upon prior literature. Our intent, however, was not to introduce entirely new categories, but instead to consolidate known attacks to provide a coherent and systematic framework for evaluating the types of harms these systems might cause. The taxonomy also introduces novel categories not considered in prior work, such as Stealing Resources.
>
> 2. We appreciate the reviewer’s concern regarding potential biases from LLM-generated data. We would like to clarify that the dataset was not generated purely by an LLM. Instead, the generation process was guided by human-annotated categories and detailed environment descriptions, supported by high-quality seed examples to ensure diversity and realism. Furthermore, each generated datapoint underwent manual inspection to verify consistency, coverage, and relevance to real-world agentic settings. This approach helped mitigate bias and enhance task realism. We will make these details clearer in the revised version.
>
> 3. We thank the reviewer for pointing out these important related works. We acknowledge that our current draft does not explicitly discuss AgentHarm. We will expand the related work section to include a detailed comparison with both benchmarks. Notably, our work differs in that it focuses on multi-agent systems, instead of the single-agent setting that was considered in these and other previous works. We have additionally included a comparison table, highlighting the differences between our benchmark and established ones, which can be found in Apendix C.
>
> 4. We thank the reviewer for highlighting this issue and we would like to rephrase this claim. Instead of considering raw-size, we compare the safety of the models with the capability in agentic systems, as measured by the quality of the plan as rated by a judge agent. Here, we found a clear correlation, where models which are able to generate high quality plans, are also less secure. These results can be found in Appendix B.1. We plan to extend these experiments in the future to all environments.
>
> 5. We would like to clarify this point. In our setup, the adversary is aware of the Guardian monitor through observation of the current trajectory and is generally instructed to act stealthily by following its original role. This means that adversaries already attempt to be covert to achieve their objectives, and the monitor’s effectiveness demonstrates its ability to detect subtle harmful behaviors even when agents are incentivized to hide them. We will clarify this aspect in the revision to better convey the robustness of our defense.
>
> 6. We thank the reviewer for emphasizing the importance of representative examples. While space constraints limited the number of examples in the main text, we have included one example per category in Appendix D to showcase the diversity and quality of tasks in our benchmark. Further, we would like to mention that the dataset will be publicly available.
>
> 7. We thank the reviewer for this suggestion. We agree that emphasizing multi-agent systems more explicitly better reflects the focus of our benchmark. In the revised version, we will update the wording throughout the paper.
>
> ### Questions
> 1. We thank the reviewer for their question. We believe our setting is plausible in real-world multi-agent scenarios. Individual agents may be compromised through existing mechanisms such as data poisoning or prompt injection attacks, and emerging research on agent marketplaces [Wang et. al, 2025] highlights scenarios where agents can be bought and sold, creating opportunities for adversarial deployment. Importantly, the experiments in this paper serve as a proof-of-concept, designed to be simple yet illustrative, to demonstrate the utility of our benchmark for evaluating robustness of agentic systems. The primary contribution of our work lies in the benchmark itself, rather than the novelty or sophistication of a particular threat, and this controlled threat setting provides a clear and consistent environment for systematic evaluation.
> However, to increase the diversity of considered threat settings, we extended our experiments to include adversaries that are able to conduct Prompt Injections in the user’s initial request (Appendix B.2.) and manipulate the output of the used tools (Appendix B.3.). We generally found that attacks still remain effective.

---

### Meta-Review · Area_Chair_V7c3 · 2026-01-05

**Summary:**

This paper introduces BAD-ACTS, a benchmark for evaluating the robustness of LLM-based multi-agent (agentic) systems under an internal adversary threat model, where one compromised agent attempts to induce other agents to execute harmful actions. It provides a taxonomy of harmful behaviors and covers five different agentic system environments. The paper finds that all tested models can be induced to execute harmful actions in the presence of an adversarial agent and proposes two simple baseline defenses.

Reviewers raised several concerns, including: (1) the novelty of the taxonomy, (2) potential biases from LLM-generated data, (3) comparison with existing benchmarks, (4) insufficient evidence for some conclusions, (5) lack of detail about the baselines, (6) the practicality of the threat model, (7) outdated literature context, (8) limitations of the evaluation metric, (9) lack of additional defense baselines, (10) missing analysis of the security–utility tradeoff, (11) missing implementation details, and (12) a lack of qualitative case analysis.

After reading the rebuttal, the Area Chair feels that the authors have partially addressed some concerns raised by the reviewers such as comparing with baselines, security-utility tradeoff, implementation details (reproducibility).  Regarding the taxonomy, which was a common concern across reviewers, the rebuttal provides reasonable clarification. The Area Chair feels that this contribution should be worded more conservatively, and that the paper should more strongly emphasize its core contribution in the multi-agent setting, rather than framing the taxonomy itself as a primary novelty.

However, several important concerns remain insufficiently addressed. For instance,

Regarding the threat model, AC believes this is a critical factor in security-related work. Reviewers raised concerns about how an adversary could realistically gain control of an agent in a multi-agent setting. While the rebuttal provides additional examples (e.g., Appendix B.2), it remains unclear how these prompt injection attacks connect concretely to reviewer's concern about realistic of the controllability

Regarding potential biases introduced by LLM-generated data, while the authors state that manual inspection was used to verify consistency, coverage, and relevance to real-world agentic settings, the rebuttal lacks sufficient detail on how this inspection was conducted, what criteria were used, and how coverage gaps or generation artifacts were identified and mitigated.

Overall, while AC appreciates the motivation and ideas behind this work, these remaining concerns are still a bit significant.

**Reviewer Concerns:**

Reviewers raised several concerns, including: (1) the novelty of the taxonomy, (2) potential biases from LLM-generated data, (3) comparison with existing benchmarks, (4) insufficient evidence for some conclusions, (5) lack of detail about the baselines, (6) the practicality of the threat model, (7) outdated literature context, (8) limitations of the evaluation metric, (9) lack of additional defense baselines, (10) missing analysis of the security–utility tradeoff, (11) missing implementation details, and (12) a lack of qualitative case analysis.

After reading the rebuttal, the Area Chair feels that the authors have partially addressed some concerns raised by the reviewers such as comparing with baselines, security-utility tradeoff, implementation details (reproducibility).  Regarding the taxonomy, which was a common concern across reviewers, the rebuttal provides reasonable clarification. The Area Chair feels that this contribution should be worded more conservatively, and that the paper should more strongly emphasize its core contribution in the multi-agent setting, rather than framing the taxonomy itself as a primary novelty.

However, several important concerns remain insufficiently addressed. For instance,

Regarding the threat model, AC believes this is a critical factor in security-related work. Reviewers raised concerns about how an adversary could realistically gain control of an agent in a multi-agent setting. While the rebuttal provides additional examples (e.g., Appendix B.2), it remains unclear how these prompt injection attacks connect concretely to reviewer's concern about realistic of the controllability

Regarding potential biases introduced by LLM-generated data, while the authors state that manual inspection was used to verify consistency, coverage, and relevance to real-world agentic settings, the rebuttal lacks sufficient detail on how this inspection was conducted, what criteria were used, and how coverage gaps or generation artifacts were identified and mitigated.

Overall, while AC appreciates the motivation and ideas behind this work, these remaining concerns are still a bit significant.

**Reviewer Scores:**

Review 867L will potentially increase the score to 4 but still will not be positive to this paper since the concerns regarding to the bias of LLM-generated data is not fully addressed.
Reviewer e46v will not increase the score since the question of the threat model is not fully addressed.
Reviewer fuFn will not increase the score since the question of the threat model is not fully addressed.
Reviewer dMap will keep the current score.

---

### Decision · Program_Chairs · 2026-01-26

Reject